# Epidemiology of revision hip replacement surgery in the UK over the past 15 years—an analysis from the National Joint Registry

Shiraz A. Sabah ,[1] Ruth Knight ,[2] Philippa J. A. Nicolson,[1] Adrian Taylor,[1] Benjamin Kendrick,[1] Abtin Alvand,[1] Stavros Petrou,[3] David J. Beard ,[1] Andrew J. Price,[1] Antony J. R. Palmer[1]

[1]Nuffield Department of Orthopaedics Rheumatology and Musculoskeletal Sciences, University of Oxford, Oxford, UK
[2]Liverpool Clinical Trials Centre, University of Liverpool, Liverpool, UK
[3]Nuffield Department of Primary Care Health Sciences, University of Oxford, Oxford, UK

**Correspondence to**
Shiraz A. Sabah;
shiraz.sabah@ndorms.ox.ac.uk

## ABSTRACT

**Objectives** To investigate trends in the incidence rate and main indication for revision hip replacement (rHR) over the past 15 years in the UK.

**Design** Repeated national cross-sectional study from 2006 to 2020.

**Setting/participants** rHR procedures were identified from the National Joint Registry for England, Wales, Northern Ireland, the Isle of Man and the States of Guernsey. Population statistics were obtained from the Office for National Statistics.

**Main outcome measures** Crude incidence rates of rHR.

**Results** The incidence rate of rHR doubled from 11 per 100 000 adults in 2006 (95% CI 10.7 to 11.3) to a peak of 22 per 100 000 adults (95% CI 22 to 23) in 2012, before falling to 17 per 100 000 adults in 2019 (95% CI 16 to 17) (24.5% decrease from peak). The incidence rate of rHR reduced by 39% in 2020 compared with 2019 (during the COVID-19 pandemic). The most frequent indications for rHR between 2006 and 2019 were loosening/lysis (27.8%), unexplained pain (15.1%) and dislocation/instability (14.7%). There were incremental increases in the annual number and incidence rates of rHR for fracture, infection, dislocation/instability and a decrease in rHR for aseptic loosening/lysis.

**Conclusions** The incidence rate of rHR doubled from 2006 to 2012, likely due to high early failure rates of metal-on-metal hip replacements. The incidence of rHR then decreased by approximately 25% from 2012 to 2019, followed by a large decrease during the COVID-19 pandemic. The decrease in the number of rHR performed for aseptic loosening/lysis may reflect improved wear and implant longevity. Increased healthcare resource will be required to care for the increasing numbers of patients undergoing rHR for fracture and infection.

## INTRODUCTION

Primary hip replacement (pHR) is a highly successful intervention and is associated with large improvements in joint function and health-related quality of life.[1] Prior to the COVID-19 pandemic, most international joint replacement registries and insurance

## STRENGTHS AND LIMITATIONS OF THIS STUDY

⇒ This study analysed a large national joint registry over a 15-year period to investigate trends in the rate of, and indication for, revision hip replacement (rHR).
⇒ Crude incidence rates were calculated using arthroplasty records from the National Joint Registry and population statistics from the Office for National Statistics.
⇒ Directly standardised rates were calculated using the 2013 European Standard Population and may facilitate international comparison of arthroplasty practices.
⇒ Registry compliance increased over the study period. In the early years of the registry, most rHR had no associated primary hip replacement record.
⇒ The analysis of re-revision procedures over-represents early modes of failure (such as infection or fracture) and under-represents late modes of failure (such as aseptic loosening/osteolysis and component wear).

databases reported increasing counts of pHR each year.[2–4] Nearly all analyses project an increase in the requirement for pHR over the coming decades,[5–7] due to ageing populations and associated increases in the prevalence of hip osteoarthritis.[8 9]

Revision hip replacement (rHR) refers to a heterogeneous group of surgical procedures that usually involve the exchange or modification of one or more components of an existing hip replacement. It is performed infrequently compared with pHR and, following a pHR most patients do not require further surgery to that hip during their lifetime.[10 11] However, increases in the number of pHR performed each year, and in younger patients, coupled with greater life expectancy, results in an increase in the number of pHR that are at risk of failure. This is expected to

create a large increase in the number of rHR procedures required in the future.[5] The epidemiology and trends in rHR are not well understood.

The National Joint Registry (NJR) for England, Wales, Northern Ireland, the Isle of Man and the States of Guernsey provides information on annual total counts of rHR.[12] NJR data have shown that the total number of rHR in the UK peaked in 2012 and has since declined. This observation is thought to be due to the high utilisation of metal-on-metal hip replacements (MOM-HR) either side of the turn of the 21st century, some of which had very high, early failure rates.[12] Count data provide only a limited perspective. They are useful for understanding the number of surgeons and surgical units performing rHR, which may, for example, inform budget estimates. However, they do not provide information on the rate of intervention, or trends in different procedures and patient groups, which requires knowledge of the size and demographics of the general population. Count data are also of limited value for comparing UK practice with other countries. It is also important to understand whether the indications for rHR have temporally changed. In revision knee replacement (rKR), the number and proportion of procedures performed for infection have increased, while procedures for aseptic loosening have decreased.[13 14] It is currently unknown whether a similar trend exists for rHR. The aim of our study was to investigate trends in the incidence rHR by indication during the past 15 years in the UK, including the impact of the COVID-19 pandemic.

## PATIENTS AND METHODS
This study was reported according to the reporting of studies conducted using observational routinely collected health data checklist.[15]

### Study data set
The NJR prospectively collects data on pHR and rHR from hospital providers within its geographic remit. Data submission has been mandatory for independent sector providers since 2003 and for NHS providers since April 2011.[16] rHR procedures are submitted to the NJR using H2 Minimum Data Set forms, which include fields on patient, surgeon, operation and implant characteristics. The current NJR definition of revision is 'any operation where one or more components are added to, removed from or modified in a joint replacement or if a Debridement And Implant Retention (DAIR) with or without modular exchange is performed'. The main change to this definition over the study period was the inclusion of DAIR procedures without modular exchange from 25 June 2018. Further information on all amendments to this definition can be found in the glossaries of NJR annual reports and the publication Operations included in the NJR.[17]

### Timeframe
This study examined NJR records for rHR performed between 1 January 2006 and 31 December 2020. We chose to exclude NJR records from 2003 to 2005, due to the known poor compliance with the registry when first formed.[18 19] Records from 2021 were excluded because: (1) the NJR data extract was produced on 3 February 2022, which meant a limited window for late submission of data; and (2) population statistics from the Office for National Statistics (ONS) were unavailable. We analysed longer term trends using records from 2006 to 2019. We hypothesised that rHR activity in 2020 was unlikely to be representative of usual practice due to the disruption in arthroplasty services as a result of the COVID-19 pandemic. To measure the impact of the COVID-19 pandemic, we compared data from 2019 (pre-pandemic) to 2020 (pandemic).

### Statistics
All statistical code is available from the lead author's GitHub page (https://github.com/shirazsabah/ox-njr-hes-ons-proms).

### Trends over time in the incidence rate of rHR
Crude incidence rates of pHR and rHR were calculated for each calendar year from 2006 to 2020. *pHR was defined as any procedure reported to the NJR using a H1 form. This included total hip replacement and hip resurfacing procedures performed for any indication, where both an acetabular and a femoral component were implanted.* The crude incidence rate was defined as the annual total count of procedures in a calendar year divided by the sum of the mid-year population estimates for adults in England, Wales and Northern Ireland. The *phe_rate* function within the PHE indicator methods package in R was used to calculate 95% CIs following Byar's method.[20 21] In some situations, a rHR may be performed as a staged procedure with two distinct operations on different days. We considered first-stage and second-stage rHR as a single procedure to calculate incidence rates. The following criteria were applied to bundle these procedures: (1) The NJR H2 form completed at the first-stage procedure was required to specify the intention to treat the patient in stages, and (2) the second-stage procedure was required to have been performed within 365 days of the first-stage. Analysis of hospital resource utilisation should count each stage individually, however, since these totals are readily available in NJR annual reports, we did not reproduce this analysis.[2] The crude incidence rate was presented as a line graph overlying bars of annual total counts. Long-term trends in total counts and incidence rates were analysed descriptively from 2006 to 2019. The percentage decrease in the total count of rHR during the COVID pandemic was analysed as the change in total count from 2019 to 2020, divided by the total count in 2019. The percentage decrease in the incidence rate of rHR was calculated following the same principles.

We investigated trends over time in the number of rHR procedures linked to a pHR on the NJR. This analysis is presented as stacked bars of annual total counts for linked and not linked procedures. Our data set did not allow us to compare incidence rates for first and subsequent rerevision procedures because the proportion of linked procedures changed markedly over the study period. For the same reason, we were unable to perform any meaningful analysis of first-linked rHR by pHR bearing surface type (eg, MOM vs other types). The incidence rate of rHR within a given year is not strongly associated with the incidence rate of pHR in that year, because few pHR fail within 1 year . The prevalence of pHR and rHR in the general population and the rate of failure of these implants is directly related to the incidence of rHR. However, we decided not to model prevalence due to the significant uncertainty around these estimates, given the longevity of hip replacements[11] and the comparatively short existence of the NJR. To enable future comparisons to populations with different age-structures (eg, from different countries around the world), we calculated directly standardised rates (DSRs) for pHR and rHR. This methodology is described in online supplemental appendix A and uses the 2013 European Standard Population.

### Trends over time in the incidence rate of rHR by patient characteristics

The study population was stratified into the following groups to investigate trends over time in the incidence rate of rHR by age at the time of revision surgery: 18–49 years/50–59 years/60–69 years/70–79 years/80+ years. We considered these groupings to fairly represent the data and to be interpretable, after exploring different categorisations. Age-specific incidence rates per 100 000 persons were calculated for each group for each year of analysis and presented graphically. Each group was further stratified into female and male to calculate age-specific and gender-specific incidence rates. Further analyses were performed to investigate trends over time in American Society of Anesthesiologists (ASA) classification and Body Mass Index (BMI).

### Trends in the main indication for rHR

Each rHR procedure was assigned a single, dominant indication for surgery using a diagnosis hierarchy based on the Australian Orthopaedic Association National Joint Replacement Registry (AOANJRR) model[22] (online supplemental appendix B table 1). Prepandemic data (2006–2019) were combined to calculate the percentage frequency of each indication. To investigate trends over time, crude incidence rates per 100 000 adults and annual percentage frequencies were calculated for each indication in each year (from 2006 to 2020, ie, including pandemic data). These data were presented using grouped barplots.

To investigate the proportional share of each revision indication in first and subsequent rerevision rHR, we analysed data from the 5 years immediately prepandemic

(2014–2019). We excluded earlier NJR data as we considered it to be most vulnerable to the maturity of the registry. We defined the following groups: first-linked rHR (the earliest revision procedure for a given patient-side linked to a primary hip replacement on the NJR); second-linked rHR (the next revision procedure for a given patient-side linked to a first-linked rHR); third or more linked rHR (subsequent revision procedure(s) linked to a second linked rHR) and no linked primary (revision procedure(s) not linked to a primary procedure). These data were presented using barplots.

### Software

Statistical analyses were performed using R V.4.2.1.

### Patient and public involvement

None.

## RESULTS

### Data cleaning and linkage

The attrition of study records during data cleaning is illustrated in the online supplemental appendix C figure 1.

### Trends over time in the incidence rate of rHR

Annual total counts and crude incidence rates of rHR procedures reported to the NJR are presented in figure 1 and online supplemental appendix D table 1. Annual total counts increased every year from 2006 (4775 rHR) to 2012 (10 217 rHR) (114% increase). Following this peak, annual total counts declined, with 8111 rHR performed in 2019 (20.6% decrease from peak). The incidence rate of rHR increased from 11.0 per 100 000 adults in 2006 (95% CI 10.7 to 11.3) to 22.3 per 100 000 adults in 2012 (95% CI 21.8 to 22.7), before falling to 16.8 per 100 000 adults in 2019 (95% CI 16.5 to 17.2) (24.5% decrease from peak). In 2020, where rHR practice was affected by the COVID-19 pandemic, 4966 rHR procedures were reported to the NJR: a crude incidence rate of 10.2 per 100 000 adults (95% CI 10.0 to 10.5). These figures represented a 38.8% reduction in the total count of rHR and a 39.0% reduction in the incidence rate of rHR in 2020 compared with 2019. The percentage of rHR procedures linked to a pHR on the NJR increased from 4.3% in 2006 to 47.5% in 2019 (rising further to 51.5% in 2020) (figure 2). The annual total counts of pHR increased every year from 2006 (48 700 pHR) to 2019 (101 828 pHR). However, increases were smaller during the last 4 years. The incidence rate of pHR increased from 112 per 100 000 adults in 2006 (95% CI 111 to 113) to 211 per 100 000 adults in 2019 (95% CI 210 to 212). This represented a 109.1% increase in the total count and 88.6% increase in the incidence rate of pHR from 2006 to 2019. In 2020, during the COVID-19 pandemic, 57 437 pHR procedures were reported to the NJR: a crude incidence rate of 119 per 100 000 adults (95% CI 118 to 120). This represented a 43.6% reduction in the total count of pHR

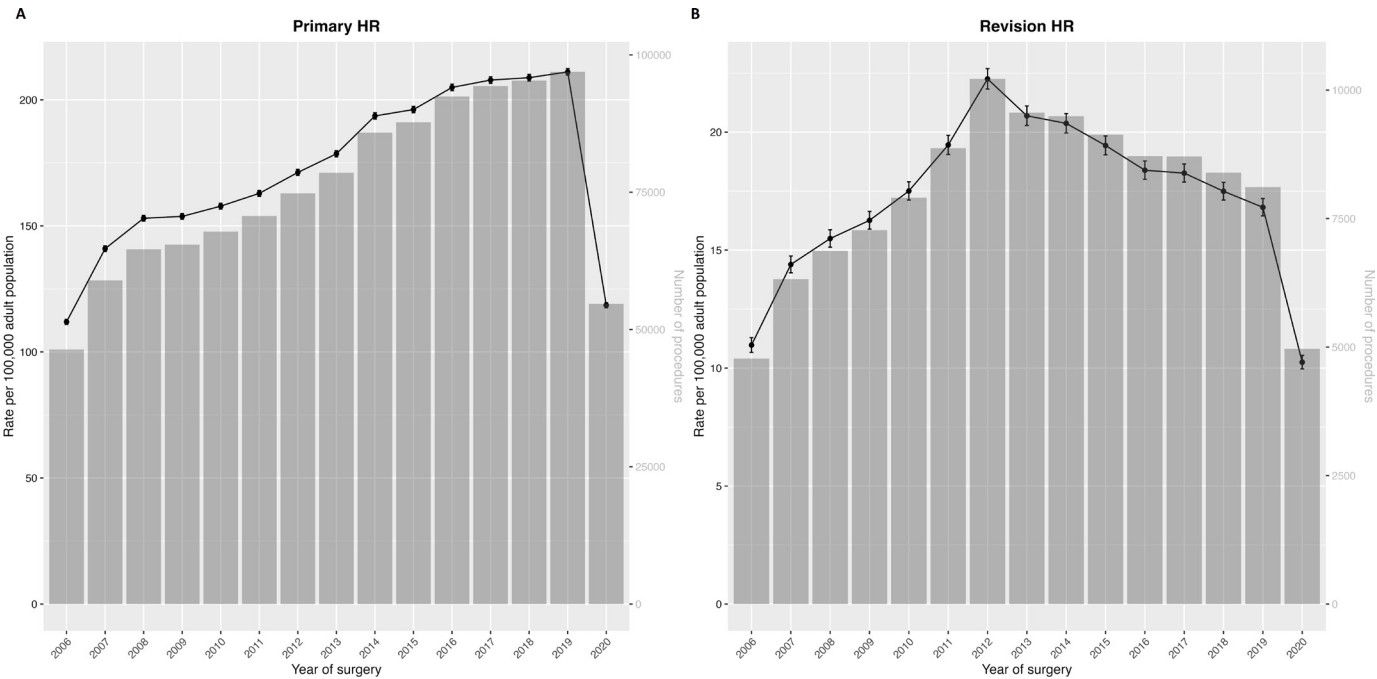

**Figure 1** Annual crude incidence rates (line, left y-axis) and total counts (bars, right y-axis) of (A) pHR and (B) rHR from 2006 to 2020. Note the differences in y-axis scales for pHR and rHR. pHR, primary hip replacement; rHR, revision hip replacement.

and a 43.8% reduction in the incidence rate of pHR in 2020 compared with 2019.

### Trends over time in the incidence rate of rHR by patient characteristics

The incidence rate of rHR increased in all age groups from 2006 to 2012 (figure 3). The group with the largest increase in the incidence rate of rHR over this period was patients aged 70–79 years at the time of surgery, from 46 per 100 000 adults in 2006 (95% CI 44 to 48) to 87 per 100 000 adults in 2012 (95% CI 85 to 90). The incidence rate of rHR declined for all age groups from 2012 to 2019, except patients aged 80 years or older at the time of surgery. In this patient group, the incidence rate of rHR peaked in 2014 (88 per 100 000 adults (95% CI 84 to 92)) and remained relatively unchanged through to 2019. Age-specific and gender-specific incidence rates for

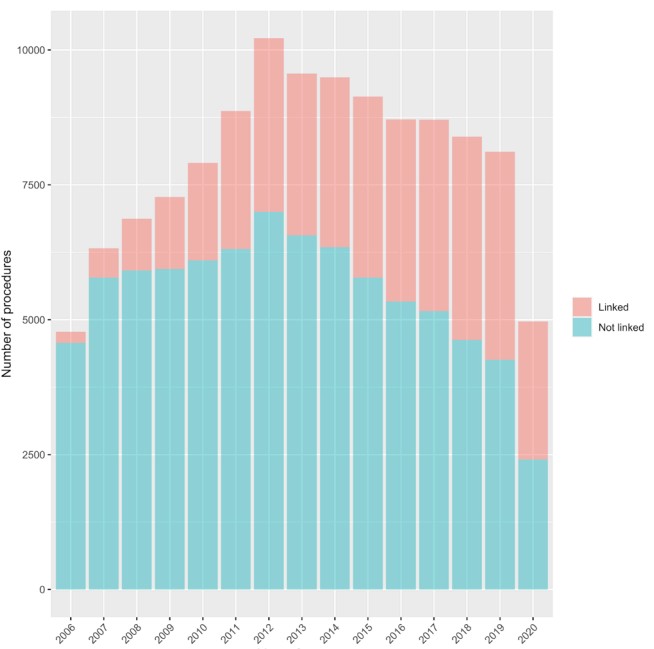

**Figure 2** Changes in the annual total counts of rHR that could be linked to a pHR on the NJR over time. NJR, National Joint Registry; pHR, primary hip replacement; rHR, revision hip replacement.

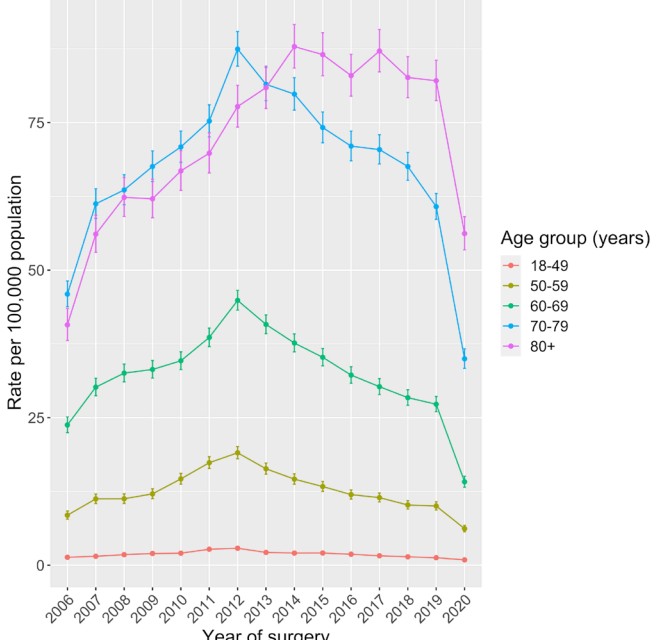

**Figure 3** Annual incidence rates of first-linked rHR from 2006 to 2020 by age group. rHR, revision hip replacement.

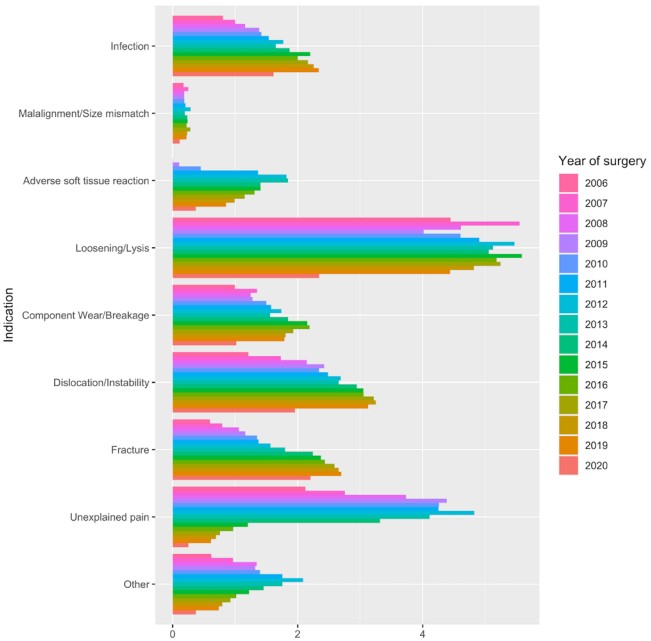

**Figure 4** Grouped barplots demonstrating annual incidence rates for all rHR by indication for surgery from 2006 to 2020. Diagnoses are ranked in hierarchical order (greatest importance at the top). rHR, revision hip replacement.

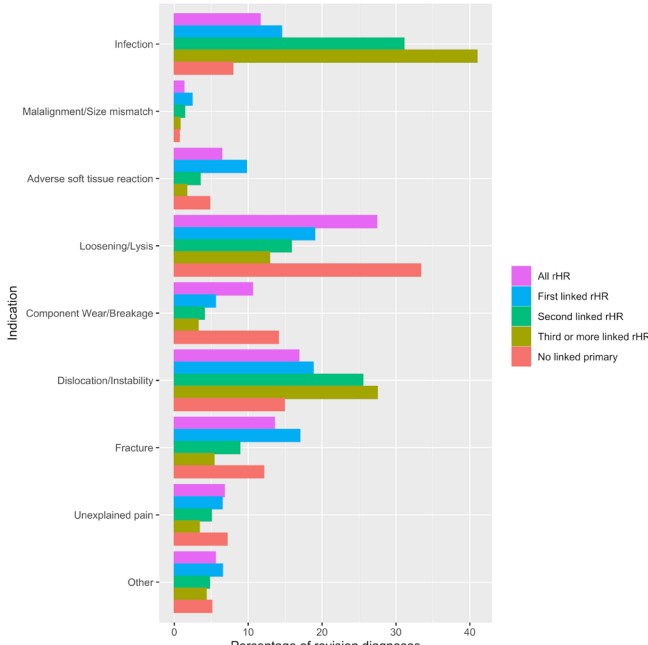

**Figure 5** Percentage frequency of each indication for first and rerevision HR from 2014 to 2019. Diagnoses are ranked in hierarchical order (greatest importance at the top). For a given procedure (eg, first-linked rHR), the sum of all diagnoses is 100%. rHR, revision hip replacement.

all rHR are provided in online supplemental appendix D figure 1. The rate of intervention was generally higher in female patients, though differences were smaller in more recent years. The analysis of patient's overall health found that the proportion of patients who were ASA class 3+ increased over the study period, from 25.3% in 2006 to 45.8% in 2019, with a further increase in 2020 (49.4%). Both ASA class 1 and ASA class 2 groups became smaller over the study period (online supplemental appendix D figure 2). The analysis of trends in BMI were difficult to interpret due to the large proportion of missing data, though this fraction reduced over the study period (online supplemental appendix D figure 3).

### Trends in the main indication for rHR

During the years 2006 to 2019 combined, aseptic loosening was the most frequent indication for rHR (27.8%), followed by unexplained pain (15.1%) and then dislocation/instability (14.7%) (online supplemental appendix B table 2). The incidence rate of rHR for infection increased nearly every year from 2006 (0.8 per 100 000 adults (95% CI 0.7 to 0.9)) to 2019 (2.3 per 100 000 adults (95% CI 2.2 to 2.5)) (figure 4). A similar trend was observed for rHR for dislocation/instability and fracture. The incidence rate for malalignment/size mismatch changed little over the study period (~0.2 per 100 000 adults). For the remaining indications, incidence rates increased to a peak before decreasing in more recent years. rHR for adverse soft tissue reaction, unexplained pain and 'other' peaked in 2012 and 2013. rHR for loosening/lysis and component wear/breakage peaked later, in 2015 and 2016, respectively.

Trends in the annual *proportional share* of each revision indication are presented in the supplementary material (online supplemental appendix D figure 4). These data should be interpreted in conjunction with crude incidence rates. For example, while the proportion of rHR for loosening/lysis was lower in 2019 compared with 2006, the crude incidence rate was greater.

The proportion of revisions attributable to each indication within first-linked rHR and rerevision rHR from 2014 to 2019 is demonstrated in figure 5. A linked pHR could not be found on the NJR for the majority of rHR performed over this period (40.1% (n=21 053/52 554) linked). Among the rHR that could not be linked to a pHR, loosening/lysis was the most frequent diagnosis (33.3%, n=10 500/31 501). Within first-linked rHR, loosening/lysis (19%, n=3448/18 138) and dislocation/instability (18.8%, n=3409/18 138)) were the most frequent diagnoses. Within linked rerevision rHR, infection was the most frequent diagnosis (31.1% s linked rHR, n=703/2263; and 41.0% third or more linked rHR, n=267/652).

### DISCUSSION

The total number and incidence rate of rHR doubled from 2006 to 2012, likely driven by high early failure rates of some types of MOM-HR. The incidence rate of rHR then fell in subsequent years, and was three-quarters of its peak by 2019, suggesting improved pHR longevity. The total number and incidence rate of pHR reported to the NJR approximately doubled from 2006 to 2019, with increases

every year until the COVID-19 pandemic. In 2020, during the COVID-19 pandemic, approximately 40% fewer rHR were performed compared with 2019. Patients aged 80 years and older displayed the largest increase in the rate of rHR during the study period. Contrary to younger age groups, where the count and rate of rHR procedures decreased after 2012, rHR activity in this older patient group was maintained. Between the years 2006 and 2019, aseptic loosening was the most frequent indication for rHR (27.8%), followed by unexplained pain (15.1%) and then dislocation/instability (14.7%). There were incremental increases in annual counts and incidence rates of rHR for infection, dislocation/instability and fracture over the study period, whereas rHR for loosening/lysis decreased in absolute and proportional terms.

Our findings differ from previous studies that have forecast progressive increases in rHR activity.[5 23 24] The peak of rHR in 2012 is likely to reflect high early failure rates of MOM-HR prostheses. Many primary MOM-HR predate the NJR. The increase in the incidence of rHR for adverse soft tissue reaction up until 2012/2013 and subsequent decrease in this indication supports this conclusion. One caveat to note that this indication was only added to the H2 MDS form in 2008 and is not exclusive to MOM-HR. Loosening/lysis was the most frequent indication for rHR overall, and for each year studied except 2009, where it was unexplained pain. We observed a decrease in the rate and proportion of rHR for loosening/lysis in the most recent years studied, which may represent low wear rates and improved longevity of modern implants. Similar to other studies, we observed large increases in rHR for infection, fracture and instability over the study period.[24]

This study is the first to provide crude incidence rates, age-specific rates and DSRs for rHR in the UK using NJR data. The strengths of this study include its reproducibility and openness. The study report was written in RMarkdown with all results derived from the source data, and statistical code made available to readers. However, the study does have limitations. Our study design is not able to differentiate between changes in reporting and changes in surgical practice. There is evidence that compliance with the NJR increased over the study period,[2 18 19] which suggests that the true decrease in rHR in recent years may be greater than our estimates. The hierarchy we have used to assign a single, dominant indication for rHR may not reflect the true diagnosis in all cases. The coding of indications for surgery was modified over the study period (eg, 'unexplained pain' was originally termed 'pain' on data collection forms). The indications reported to the NJR reflect those thought to apply at the time of surgery. In the case of infection, which is sometimes diagnosed postoperatively, there is evidence to suggest it may be under-recognised by registries.[25] Procedures that are not considered a revision procedure—for example, operative fixation of a periprosthetic fracture, were not included in the study. There is continued uncertainty whether periprosthetic hip fractures should be managed with revision or fixation, and trends have changed over time.[26 27] Our

subgroup analysis of first-linked, second-linked and third or more-linked rHR excluded procedures with no pHR recorded in the registry. It is likely that our analyses of rerevision procedures overrepresent failure modes occurring in early follow-up (such as infection or fracture) and under-represent failure modes more likely to occur at later timepoints (such as aseptic loosening/osteolysis and component wear).[28] We attempted to mitigate this effect by restricting our analysis to more mature registry data (2014 to 2019), but longer term data are required. We have not projected future requirements for rHR.

Our study demonstrates the considerable disruption to rHR practice during the COVID-19 pandemic, similar to trends recently reported for pHR, pKR and rKR.[14 29] In the near future, the number of rHR performed is likely to increase to fulfil the backlog of patients whose care was delayed during the COVID-19 pandemic.[30] It is important to consider that the smaller than expected increases in rHR may be due to limited capacity in the healthcare system to deliver more of these procedures. There are currently major infrastructure projects in the UK to increase capacity for both primary and revision joint replacement procedures. These include development of surgical hubs to separate elective and emergency care, with the idea that planned surgical activity may continue with fewer disruptions during surges of unplanned hospital admissions.[31] Over the next few years, it will be important to measure the effectiveness of these programmes. Our study has also investigated trends over time in the main indication for rHR. The recent decreases we observed in rHR for loosening/lysis and adverse soft tissue reaction may reflect greater use of highly rated implants.[32] Similarly, decreases in rHR for 'unexplained pain' may suggest that evidence-based indications for revision surgery are being followed. These may mitigate the future revision burden. In contrast, if current trends continue, fracture, infection and instability will represent an increasing proportion of rHR procedures. This has implications for healthcare provision since these indications are associated with greater complication rates, costs and utilisation of hospital resources in comparison to elective, aseptic revision joint replacement procedures.[33–36]

In conclusion, the incidence rate of rHR doubled from 2006 to 2012, likely due to high early failure rates of MOM-HR. The incidence of rHR decreased by approximately 25% from 2012 to 2019, likely due to increased implant longevity. Recent trends suggest a rise in rHR for fracture, infection and dislocation/instability, and a fall in rHR for loosening/lysis. There was a large reduction in the incidence of rHR during the COVID-19 pandemic in 2020.

**Acknowledgements** We thank the patients and staff of all the hospitals who have contributed data to the National Joint Registry. We are grateful to the Healthcare Quality Improvement Partnership (HQIP), the NJR Research Committee and staff at the NJR for facilitating this work. The authors have conformed to the NJR's standard protocol for data access and publication.

**Contributors**  All authors made substantial contributions to the work. CRediT roles: SAS: conceptualisation, data curation, formal analysis, funding acquisition, investigation, methodology, software, writing—original draft, guarantor. RK: data curation, formal analysis, statistical expertise, investigation, methodology, software, writing—review and editing. PJAN: investigation, methodology, software, writing—review and editing. AT, BK, AA, SP: investigation, writing—review and editing. DB and AJPr: conceptualisation, funding acquisition, investigation, methodology, supervision, writing—review and editing. AJPa: conceptualisation, investigation, methodology, supervision, writing—review and editing.

**Funding**  This study received funding from the National Institute of Health Research (Doctoral Research Fellowship 301771).

**Disclaimer**  The views expressed represent those of the authors and do not necessarily reflect those of the National Joint Registry Steering Committee, Research Sub-committee or the Healthcare Quality Improvement Partnership (HQIP) who do not vouch for how the information is presented.

**Competing interests**  SAS is funded by an NIHR Doctoral Research Fellowship, Royal College of Surgeons One Year Fellowship and funding from the Rosetrees Trust, and is a member of the editorial board of the Bone & Joint Journal. AA has received funding from Zimmer Biomet and Medacta International outside of the submitted work. AJP has received funding from Zimmer Biomet outside of the submitted work and is a board member of the British Orthopaedic Association and British Association for Surgery of the Knee. BK has received funding from Adler Ortho outside of the submitted work. SP is an NIHR Senior Investigator. DB has received institutional funding from the NIHR and Rosetrees Trust.

**Patient and public involvement**  Patients and/or the public were not involved in the design, or conduct, or reporting or dissemination plans of this research.

**Patient consent for publication**  Not required.

**Ethics approval**  Ethical approval was obtained from the London-Bromley Research Ethics Committee (20/LO/0428). Data access approvals were obtained from the National Joint Registry (NJR) for England, Wales, Northern Ireland, the Isle of Man and the States of Guernsey (RSC2017/26). Data for patients who did not to consent to the NJR audit were not included.

**Provenance and peer review**  Not commissioned; externally peer reviewed.

**Data availability statement**  Data may be obtained from a third party and are not publicly available. Data are available from the National Joint Registry following approval by their Research Subcommittee.

**ORCID iDs**
Shiraz A. Sabah http://orcid.org/0000-0003-2401-1372
Ruth Knight http://orcid.org/0000-0001-6810-2845
David J. Beard http://orcid.org/0000-0001-7884-6389

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
