## [Reviewer comments · BMJ Open]

ARTICLE DETAILS

TITLE (PROVISIONAL)	Epidemiology of revision hip replacement surgery in the UK over the past 15 years - An analysis from the National Joint Registry
AUTHORS	Sabah, Shiraz; Knight, Ruth; Nicolson, Philippa; Taylor, Adrian; Kendrick, Benjamin; Alvand, Abtin; Petrou, Stavros; Beard, David; Price, Andrew; Palmer, Antony

VERSION 1 – REVIEW

REVIEWER	Willem Schreurs Radboudumc Instituut voor Wetenschappelijk Onderwijs en Opleidingen, Orthopedics
REVIEW RETURNED	27-Feb-2023

GENERAL COMMENTS	Abstract: No comments Introduction: No comments Patients and Methods: I am not a statistician but I believe, given the reputation of the group, that the used methods are appropriate. indeed, it is a problem that the authors do not know whether a revision is a first, second or even third revision. However, this does not influence the revision load of the country. Of course, these figures get better with a more mature registry, linkage of the pHR to the rHR is then possible. Results: it is remarkable that the over 80 group has a persisting rHR, probably indicating that is the older patients not the best implants are used (more and more uncemented and in fact it is common sense that cemented implants in over 75 years should be the gold standard). This is probably also causing the in time still raising number of rHR for fractures (fig 4). Discussion: No comments.
--

REVIEWER	Thomas Wainwright Bournemouth University, Bournemouth University
REVIEW RETURNED	17-Mar-2023

GENERAL COMMENTS	Thank you for the opportunity to review this very well written report. The analysis method is clearly stated and easy to follow, and the conclusions are balanced. However, some points below that I think the authors should consider/address within the text. The authors attribute the increase in rHR rate 2006-2012 to be likely
--

	due to failure rates of metal-on-metal hip implants. I agree with this likelihood, but I wonder why they did not analyse the trends over time in the incidence rate of rHR by linked primary bearing surface combination. It would be useful to provide clarification why this was not done or if it was not methodologically possible. The authors also conclude that the decrease in the number of rHR performed for aseptic loosening/lysis may reflect improved properties of modern implants. Again, this a sensible conclusion, but I also wonder if they considered looking at the trends over time in the incidence rate of rHR by the linked primary method of fixation. For example, uncemented cup use has increased. Again, it would be useful to provide clarification why this was not done or if it was not methodologically possible. These parameters are collected within the NJR, so one may presume the data was available.
--	--

REVIEWER	Bart Pijls Leiden Universitair Medisch Centrum
REVIEW RETURNED	11-Jun-2023

GENERAL COMMENTS	The authors present an epidemiological study on the annual incidence rates for – as they call it – revision hip replacement in the UK during the period 2006 to 2020. While this is certainly an interesting and well conducted study there are some points that need attention. Please see my comments below. It would be interesting to include up-to-date data, so up to and including 2022 as this would also give some insight into the recovery after COVID-19. Please define more clearly what is meant by primary hip replacement. Does this include hemi-arthroplasties for e.g. fractures or is it restricted to primary total hip replacement? Please define more clearly what is meant by revision hip replacement. Is this restricted to revision for parts that are fixed to the bone (femoral stem and/or acetabular cup) or does this include minor revisions of modular components only such as modular heads? What were indications of primary hip replacement were considered? All indications including fractures and tumour or was this restricted to primary osteoarthritis? It would be interesting to also see the numbers of primary hip replacement at risk of failure. The authors need to be more convincing why the incidence rate for revision hip replacement is relevant as compared to e.g. the number of primary hip replacements at risk. Regarding the incidence rates of revision hip replacement, an analysis stratifying for metal-on-metal revisions and non metal-on-metal revisions would be very informative. While the age-specific incidence rates are highly informative, age is not the only prognostic factor for revision. Hence a stratified analyses for gender, BMI and ASA-grade would be advisable.
--

	The discussion is well balanced and the conclusion is supported by the results.
--	---

REVIEWER	Daniel Gould The University of Melbourne, Department of Surgery
REVIEW RETURNED	14-Jul-2023

GENERAL COMMENTS	Thank you for the opportunity to review this manuscript. The authors have presented a clear and high-quality report of an analysis of a pertinent topic using a rich dataset. The trends over time for revision hip arthroplasty are extremely valuable pieces of information for public health, health economics, and orthopaedic clinical practice. I believe the analysis was clearly described and rigorous, and the Conclusions support the findings. Furthermore, the Discussion section provides interesting and relevant context for the interpretation of these findings. I would especially like to commend the authors on their acknowledgement of the key limitations in their study, as this is essential information to ensure accurate and appropriate interpretation of findings, particularly regarding the use of revision hip arthroplasty procedures for which there was a linked primary procedure in the NJR. My comments and suggests are provided to help improve the quality of certain aspects of the manuscript such that the strengths of this excellent paper are even clearer to readers.  1. Page 12, lines 186-188: suggest providing reference to the Appendix, where it is explained why these age brackets were selected 2. Page 21, lines 332-334: suggest providing a reference for the rate of these longer-term complications, from prior literature. This would pair nicely with the final paragraph of the Results section in which the proportion of non-linked rHRs due to loosening/lysis is specified. Expanding upon this, there may be an opportunity to somewhat re-frame the title and/or abstract of the paper to make it clearer, from the outset, that the focus of this paper is on short(er) term complications because this is what can be reliably analysed using this comprehensive and expanding registry. 3. Page 21, line 342: suggest expanding upon the 'implications for healthcare provision'. I agree with this statement, and I do not believe the authors need to go into great detail, but a brief summary of some of the main implications would, in my opinion, strengthen the Discussion section. 4. Not all of the figures appear to have error bars. Perhaps this was intentional, and I do not believe it is a major issue, but just in case there were error bars omitted erroneously, I felt it should be mentioned.
---

REVIEWER	Erica Sercy Henry M Jackson Foundation for the Advancement of Military Medicine Inc
REVIEW RETURNED	19-Jul-2023

GENERAL COMMENTS	This manuscript reports on calculation of crude incidence, standardized incidence rates, and indications for revision hip replacement using data from the National Joint Registry. The study was well designed and is described thoroughly and clearly in the manuscript. I have very few comments.
---

	I assume that in using registry data from the NJR, you were exempt from ethical/Institutional Review Board approval; if that is the case, please just briefly state that in the Methods section. I especially appreciate that you addressed not knowing whether the changes and trends observed in the data were due to changes in reporting or in actual practice. I think that's an important point to acknowledge, especially as the data source (NJR registry) was relatively new in your earlier study timepoints and may not have had great reporting adherence. Everything in the manuscript was clear, and I commend the authors on a study that was designed and implemented well. My only suggestion would be a bit more discussion as to the impact or relevance of the results. How do you anticipate they might be used to influence clinical practice? For example, in your analyses of indication for revision, you note changes over time, such as revision because of infection increasing over time. Do the resources needed for the revision change according to the reason for the revision? That is, does it matter why the revision is being done, or is the revision process the same regardless? Or perhaps, are some indications more likely to need additional revisions in the future (per the literature), e.g., if, for example, revisions due to infection are more likely to need a second or third revision, you might suggest that the increase in infection indication seen recently may potentially lead to more additional revisions in the future (or some other similar reasoning regarding indication). Essentially, why does it matter to report on reasons for revisions - can knowing the reasons somehow be used in attempts to avert need for revision? Another example that could have used more discussion on implications is that you actually saw a decrease in revision rates after a peak in 2012, likely due to better materials used in the newer hip replacements - what kind of implications might this decrease have? I know you don't want to forecast or predict trends into the future, but I am left with a slight feeling that I'm not sure what the real-world impact of the study might be or what drove you to look into this question. Thanks for the great work.
--	--

VERSION 1 – AUTHOR RESPONSE

Reviewer 1: Dr. Willem Schreurs, Radboudumc Instituut voor Wetenschappelijk Onderwijs en Opleidingen

Reviewer comment	Abstract: No comments Introduction: No comments Patients and Methods: I am not a statistician but I believe, given the reputation of the group, that the used methods are appropriate. indeed, it is a problem that the authors do not know whether a revision is a first, second or even third revision. However, this does not influence the revision load of the country. Of course, these figures get better with a more mature registry, linkage of the pHR to the rHR is then possible. Results: it is remarkable that the over 80 group has a persisting rHR, probably indicating that is the older patients not the best implants are used (more and more
---

	uncemented and in fact it is common sense that cemented implants in over 75 years should be the gold standard). This is probably also causing the in time still raising number of rHR for fractures (fig 4). Discussion: No comments.
Author response	Thank you for your review. We were also interested to observe the growth in revision procedures for patients 80 years and older. This is likely to represent an ageing population, greater numbers of joint replacements at risk, and widening of candidacy for revision surgery.
Author action	None required.

Reviewer 2: Prof. Thomas Wainwright, Bournemouth University

Reviewer comment	Thank you for the opportunity to review this very well written report. The analysis method is clearly stated and easy to follow, and the conclusions are balanced. However, some points below that I think the authors should consider/address within the text. The authors attribute the increase in rHR rate 2006-2012 to be likely due to failure rates of metal-on-metal hip implants. I agree with this likelihood, but I wonder why they did not analyse the trends over time in the incidence rate of rHR by linked primary bearing surface combination. It would be useful to provide clarification why this was not done or if it was not methodologically possible.
Author response	Thank you for your review. We were also very interested to explore this trend. The investigation of trends over time in the incidence rate of rHR by primary bearing surface combination requires that the rHR be linked to a pHR on the NJR. In the original submission, we decided that the data available would not provide a meaningful analysis due to the low linkage rate of rHR to pHR in the early years of the NJR (as illustrated in Figure 2). In response to reviewer comments, we did review our position and perform some exploratory work. However, ~80% of rHR in the years 2006-2012 were not linked to a pHR. We felt that trend analysis in the context of this much 'missing' data was likely to be misleading.
Author action	No changes made.
Reviewer comment	The authors also conclude that the decrease in the number of rHR performed for aseptic loosening/lysis may reflect improved properties of modern implants. Again, this a sensible conclusion, but I also wonder if they considered looking at the trends over time in the incidence rate of rHR by the linked primary method of fixation. For example, uncemented cup use has increased. Again, it would be useful to provide clarification why this was not done or if it was not methodologically possible. These parameters are collected within the NJR, so one may presume the data was available.
Author response	Thank you for your comment. We agree that this question (and many like it – for instance, the influence of cup

	fixation, stem fixation, bearing surface combinations, brands, etc.) are of interest. Our comments above regarding the ability to analyse pHR bearing surface combination also apply here. We also considered these research questions to relate more directly to the survivorship of the pHR, rather than to the incidence of rHR (because it depends how many of each type of procedure were performed and their hazard function). These questions require a different set of methodologies (i.e. survival analysis with risk adjustment and investigation of trends over time). For interest, the NJR do provide a nice example of an analysis of trends over time in the survivorship of pKR – see the ‘landmark plot’ in the NJR 17th Annual Report page 131 Fig 3.K3. We agree that there is more to do in this area.
Author action	None required

Reviewer 3: Dr. Bart Pijls, Leiden Universitair Medisch Centrum

Reviewer comment	The authors present an epidemiological study on the annual incidence rates for – as they call it – revision hip replacement in the UK during the period 2006 to 2020. While this is certainly an interesting and well conducted study there are some points that need attention. Please see my comments below. It would be interesting to include up-to-date data, so up to and including 2022 as this would also give some insight into the recovery after COVID-19.
Author response	Thank you for your review. We agree that analysis of more contemporary data would be of interest. Unfortunately, these data are not currently available to us (nor likely to other research groups). The data access process for the current study took over 2 years with applications to a research ethics committee, NJR, NHS Digital and the Confidentiality Advisory Group. We also recommend a ‘grace’ period to allow for late submission of records by hospitals. Finally, to create standardised rates requires population data from the Office for National Statistics. At the time of submission (>6 months ago), we were using the most recent data.
Author action	None required.
Reviewer comment	Please define more clearly what is meant by primary hip replacement. Does this include hemi-arthroplasties for e.g. fractures or is it restricted to primary total hip replacement?
Author response	We included all procedures submitted on NJR H1 forms. Over the study period, this included any procedure (for any indication) where both an acetabular component and a femoral component were implanted. Newer H1 forms (released this year) do include hip hemiarthroplasties, but we have not included these. We do not believe that further analysis of trends in characteristics of pHR fall within the scope of this study.
Author action	We have included the following definition: “pHR was defined as any procedure reported to the NJR using a H1 form. This included total hip replacement and hip resurfacing procedures performed for any indication, where both an acetabular and a femoral component were implanted.”

Reviewer comment	Please define more clearly what is meant by revision hip replacement. Is this restricted to revision for parts that are fixed to the bone (femoral stem and/or acetabular cup) or does this include minor revisions of modular components only such as modular heads?
Author response	The NJR definition includes both 'major' and 'minor' revision hip replacement procedures. We acknowledge that some readers may be interested in trends in 'major' or 'minor' rHR over time. However, we have generally tried to avoid these distinctions. From the perspective of the patient, both categories constitute 'an operation'. We acknowledge that, from the perspective of the healthcare system, there are differences in resource utilisation.
Author action	Note definition provided: The current NJR definition of revision is "any operation where one or more components are added to, removed from or modified in a joint replacement or if a Debridement And Implant Retention (DAIR) with or without modular exchange is performed".
Reviewer comment	What were indications of primary hip replacement were considered? All indications including fractures and tumour or was this restricted to primary osteoarthritis?
Author response	We considered all indications for pHR.
Author action	None required.
Reviewer comment	It would be interesting to also see the numbers of primary hip replacement at risk of failure. The authors need to be more convincing why the incidence rate for revision hip replacement is relevant as compared to e.g. the number of primary hip replacements at risk.
Author response	We agree that the prevalence of hip replacements in the population and their rate of failure (hazard function) has a direct relationship to the rate of revision hip replacement (assuming the system has capacity to fulfil these operations). We discussed this in the original submission and have made some changes to this section in the revision. We did consider modelling prevalence. However, we felt that uncertainty around estimates based on NJR data would be very large. Many assumptions would be necessary, for example, around:  - The incidence of primary and revision hip replacement prior to, and during the early years of, the NJR; - The failure rate of primary and revision hip replacements prior to, and during the early years of, the NJR. There may be other datasets in the UK (going back further in time) better suited to answer this question. For reference, the following study in the US estimated prevalence of hip and knee replacement. Key to this study was the availability of data over a long period, starting from the inception of hip and knee replacement: Maradit Kremers H, Larson DR, Crowson CS, et al. Prevalence of Total Hip and Knee Replacement in the United States. The Journal of bone and joint surgery American volume. 2015;97(17):1386-1397. doi:10.2106/JBJS.N.01141 Our rationale for including the incidence rate of pHR was to provide some additional context for readers by which to frame the 'commonness' of rHR procedures.

Author action	“The prevalence of pHR and rHR in the general population and the rate of failure of these implants is directly related to the incidence of rHR. However, we decided not to model prevalence due to the significant uncertainty around these estimates, given the longevity of hip replacements [11] and the comparatively short existence of the NJR.”
Reviewer comment	Regarding the incidence rates of revision hip replacement, an analysis stratifying for metal-on-metal revisions and non metal-on-metal revisions would be very informative.
Author response	Please see comments to Reviewer #2.
Author action	No changes.
Reviewer comment	While the age-specific incidence rates are highly informative, age is not the only prognostic factor for revision. Hence a stratified analyses for gender, BMI and ASA-grade would be advisable.
Author response	Thank you. We have included these analyses.
Author action	See additions to Methods, Results and Appendix D.
Reviewer comment	The discussion is well balanced and the conclusion is supported by the results.
Author response	Thank you.
Author action	None required.

Reviewer 4: Mr. Daniel Gould, The University of Melbourne

Reviewer comment	Thank you for the opportunity to review this manuscript. The authors have presented a clear and high-quality report of an analysis of a pertinent topic using a rich dataset. The trends over time for revision hip arthroplasty are extremely valuable pieces of information for public health, health economics, and orthopaedic clinical practice. I believe the analysis was clearly described and rigorous, and the Conclusions support the findings. Furthermore, the Discussion section provides interesting and relevant context for the interpretation of these findings. I would especially like to commend the authors on their acknowledgement of the key limitations in their study, as this is essential information to ensure accurate and appropriate interpretation of findings, particularly regarding the use of revision hip arthroplasty procedures for which there was a linked primary procedure in the NJR.
Author response	Thank you for your comments.
Author action	None required.
Reviewer comment	My comments and suggests are provided to help improve the quality of certain aspects of the manuscript such that the strengths of this excellent paper are even clearer to readers. 1. Page 12, lines 186-188: suggest providing reference to the Appendix, where it is explained why these age brackets were selected
Author response	Thank you. We agree that age-specific rates are sensitive to how the data are categorised. We explored several different age (and sex) categorisations and considered the groups chosen to fairly represent the data. These groups are also easily interpreted by

	readers. We have favoured 'interpretability' over alternative (mathematical) approaches of cutting the data (such as dividing the sample into equally sized groups). The appendix provides information on the direct standardisation work we performed (which is a different process to the calculation of age-specific rates).
Author action	None required.
Reviewer comment	2. Page 21, lines 332-334: suggest providing a reference for the rate of these longer-term complications, from prior literature. This would pair nicely with the final paragraph of the Results section in which the proportion of non-linked rHRs due to loosening/lysis is specified. Expanding upon this, there may be an opportunity to somewhat re-frame the title and/or abstract of the paper to make it clearer, from the outset, that the focus of this paper is on short(er) term complications because this is what can be reliably analysed using this comprehensive and expanding registry.
Author response	Thank you. We agree this is beneficial and have included a reference to a review that discusses modes of failure for THR over time. We were unable to find registry reports on the cumulative incidence of revision by diagnosis or the hazard by diagnosis at different time points. The NJR reports only PTIR estimates of indications for revision. Regarding the second point (related to shorter term complications), our preference would be to leave the title and abstract unchanged. We have looked at the epidemiology of rHR over a 15-year period, and most of the analyses reflect this (e.g. Figs 1-4). For example, if we select out Figure 4 for attention, this demonstrates changes in the indications for all rHR over the 15-year period (irrespective of whether they could be linked to a pHR). It is only the re-revision work (e.g. Fig 5) that is sensitive to failure modes occurring earlier. The registry will represent a lifetime cohort for more and more patients as time goes on and become less sensitive to this phenomenon.
Author action	We have made a small change to this sentence and added a reference: "It is likely that our analyses of re-revision procedures over-represent failure modes occurring in early follow-up (such as infection or fracture) and under-represent failure modes more likely to occur at later timepoints (such as aseptic loosening/osteolysis and component wear) [28]."
Reviewer comment	3. Page 21, line 342: suggest expanding upon the 'implications for healthcare provision'. I agree with this statement, and I do not believe the authors need to go into great detail, but a brief summary of some of the main implications would, in my opinion, strengthen the Discussion section.
Author response	Thank you. We agree this needed further clarification. We have expanded the Discussion section (L376-395). Please also see comments to Reviewer #5.
Author action	As above.
Reviewer comment	4. Not all of the figures appear to have error bars. Perhaps this was intentional, and I do not believe it is a major issue, but just in case there were error bars omitted erroneously, I felt it should be mentioned.
Author response	Thank you. This is an excellent point! - Fig 1a and Fig 1b – 95% CIs were missing and have been added.

	 - Fig 3 and App B Fig 1 – 95% CIs already present, but note for revisions these are almost 'swallowed' within the data point at the size/resolution supplied. - We decided not to include 95% CIs for barplots (e.g. Fig 2; Fig 4; Fig 5) as these make them visually less appealing.
Author action	As above.

Reviewer 5: Ms. Erica Sercy, Henry M Jackson Foundation for the Advancement of Military Medicine Inc

Reviewer comment	This manuscript reports on calculation of crude incidence, standardized incidence rates, and indications for revision hip replacement using data from the National Joint Registry. The study was well designed and is described thoroughly and clearly in the manuscript. I have very few comments. I assume that in using registry data from the NJR, you were exempt from ethical/Institutional Review Board approval; if that is the case, please just briefly state that in the Methods section.
Author response	Thank you for your review. This work required data access approval from the NJR (RSC2017/26) and research ethics approval (20/LO/0428). This information is provided in the 'Ethics approval statement'. I expect this may have been redacted from the review copy.
Author action	None required.
Reviewer comment	I especially appreciate that you addressed not knowing whether the changes and trends observed in the data were due to changes in reporting or in actual practice. I think that's an important point to acknowledge, especially as the data source (NJR registry) was relatively new in your earlier study timepoints and may not have had great reporting adherence.
Author response	Thank you.
Author action	None required.
Reviewer comment	Everything in the manuscript was clear, and I commend the authors on a study that was designed and implemented well. My only suggestion would be a bit more discussion as to the impact or relevance of the results. How do you anticipate they might be used to influence clinical practice? For example, in your analyses of indication for revision, you note changes over time, such as revision because of infection increasing over time. Do the resources needed for the revision change according to the reason for the revision? That is, does it matter why the revision is being done, or is the revision process the same regardless? Or perhaps, are some indications more likely to need additional revisions in the future (per the literature), e.g., if, for example, revisions due to infection are more likely to need a second or third revision, you might suggest that the increase in infection indication seen recently may potentially lead to more additional revisions in the future (or some other similar reasoning regarding indication). Essentially, why does it matter to report on reasons for revisions - can knowing the reasons somehow be used in attempts to avert need for revision?
Author response	Thank you. We have recently completed a study to investigate the relationship of the indication for surgery and patient-relevant outcomes following first rHR. We found that the indication for revision joint replacement was an important predictor of patient-

	relevant outcomes (re-revision, mortality, serious medical complications, PROMs and length of hospital stay). In general, 'urgent' indications were associated to greater cost and resource utilisation. Unfortunately, our hip study is undergoing peer review and not yet available to be cited.
Author action	See changes to Discussion (L376-L395).
Reviewer comment	Another example that could have used more discussion on implications is that you actually saw a decrease in revision rates after a peak in 2012, likely due to better materials used in the newer hip replacements - what kind of implications might this decrease have? I know you don't want to forecast or predict trends into the future, but I am left with a slight feeling that I'm not sure what the real-world impact of the study might be or what drove you to look into this question. Thanks for the great work.
Author response	Thank you for these comments. We expect our findings will come as a surprise to many readers – especially those who have read forecasts of exponential increases in revision hip replacement. We agree that more discussion was needed around the implications of our study findings and have expanded the Discussion section accordingly.
Author action	See changes to Discussion (L376-L395).

VERSION 2 – REVIEW

REVIEWER	Willem Schreurs Radboudumc Instituut voor Wetenschappelijk Onderwijs en Opleidingen, Orthopedics
REVIEW RETURNED	22-Aug-2023
GENERAL COMMENTS	Have seen the answers to all reviewers. I think the authors adressed all well, speaking for my part I was already satisfied after my previous review. Therefore I attached no further comments or files
REVIEWER	Bart Pijls Leiden Universitair Medisch Centrum
REVIEW RETURNED	20-Aug-2023
GENERAL COMMENTS	Thank you for these revisions. The authors have addressed my comments in a satisfactory manner.
REVIEWER	Daniel Gould The University of Melbourne, Department of Surgery
REVIEW RETURNED	31-Aug-2023
GENERAL COMMENTS	I would like to thank the authors for thoughtfully and thoroughly addressing my comments, and those of other reviewers. I have no

	further comments and I believe this manuscript is a valuable contribution to the literature.
REVIEWER	Erica Sercy Henry M Jackson Foundation for the Advancement of Military Medicine Inc
REVIEW RETURNED	17-Aug-2023
GENERAL COMMENTS	Thank you for addressing my comments and adding to the discussion section. I have no further questions or recommendations for this manuscript. Again, commendations on a well-designed and well-reported study.